# OpenReview forum: "Turning Up the Heat: Min-p Sampling for Creative and Coherent LLM Outputs"
_ICLR.cc/2025/Conference — ICLR 2025 Oral_

### Official Review · Reviewer_NZFq · 2024-10-28

**Soundness:** 3
**Presentation:** 3
**Contribution:** 3
**Rating:** 6
**Confidence:** 4

**Summary:**

The paper introduces a novel dynamic truncation method called min-p sampling, which adeptly adjusts the sampling threshold based on the model’s confidence by scaling according to the top token’s probability. This approach presents a significant advancement over traditional sampling methods like top-p and top-k, demonstrating improved balance between the quality and diversity of generated text.
The authors conducted experiments across three datasets, yielding compelling results that underscore the effectiveness of min-p sampling. The findings indicate that this method not only enhances the quality of text generation but also fosters greater diversity, which is a critical aspect in natural language processing tasks.
The writing in this paper is clear and accessible, making the concepts relatively easy to understand. The methodology is straightforward and provides a meaningful contribution to the field. Overall, this paper presents insights and a potential solution to the challenges of text generation, which may be of interest to researchers and practitioners.

**Strengths:**

1. The paper is well-written and easy to follow.
2. The proposed min-p sampling makes an effective balance between coherence and diversity in text generation.

**Weaknesses:**

Please refer to Questions.

**Questions:**

Q1: In Table 2, the experimental results on the GPQA Main and GSM8K datasets demonstrate that Min-p sampling achieves better accuracy compared to other sampling methods when the temperature is set to 1 or higher. Additionally, it appears that all sampling methods perform better at lower temperature values.

We are particularly interested in the ceiling performance of these methods on these two datasets. However, when the temperature is set to 0.7, min-p sampling does not show a significant advantage over top-p sampling. If the temperature is further decreased (e.g., to 0.5 or 0.3), will the performance of top-p sampling continue to improve? Furthermore, does min-p sampling still maintain a significant advantage over top-p sampling at these lower temperature settings?

Q2: Figure 1 shows that top-p sampling can ensure diversity in generation but may result in incoherent content. On the other hand, top-k sampling can ensure generation of high probability text but may lose diversity. Can a combination of top-p and top-k sampling compensate for their respective shortcomings and better balance coherence and diversity? Would min-p sampling be more effective than the combined method of top-p and top-k sampling?

---

> ### Author Response · Authors · 2024-11-22
> **Response to Reviewer NZFq**
>
> We thank the reviewer for their detailed and thoughtful feedback. Your insights have been invaluable in strengthening our work. Below, we address your specific questions with additional experiments and analyses.
>
> ---
>
> ## Responses to Questions
>
> ### **Q1: Performance at Lower Temperatures**
>
> Thank you for raising this important question. Based on your comment, we conducted additional evaluations on GPQA and GSM8K datasets at lower temperature settings (0.0–0.5) and have included the results in Appendix D.2. Here is a summary:
>
> 1. **Convergence at Low Temperatures:**
>    - At temperatures close to 0, all sampling methods—including min-p, top-p, and temperature-only—produce nearly identical outputs due to the sharp peak in the token distribution. This convergence results in less than a 1% difference in performance among the methods.
>
> 2. **Performance Divergence Below 1.0:**
>    - As temperature increases slightly (e.g., 0.3–0.5), top-p and min-p begin to diverge. However, the differences remain within a 1% range, as the token distribution remains highly peaked, limiting the diversity introduced by sampling.
>
> 3. **Key Observations:**
>    - At temperatures below 1.0, min-p maintains slight but consistent advantages in coherence and diversity due to its dynamic thresholding.
>    - Top-p sometimes shows marginally higher accuracy at specific settings (e.g., top-p = 0.9 at temperature = 0.5 on GPQA), but the differences are not statistically significant.
>
> **Conclusion:**
> At very low temperatures, the performance differences among sampling methods are minimal. Min-p's advantages become more pronounced as the temperature increases beyond 1.0, where its adaptive truncation better balances coherence and diversity.
>
> ---
>
> ### **Q2: Combining Top-p and Top-k Sampling**
>
> We tested combinations of top-p and top-k sampling across a range of parameter settings and evaluated their performance on GPQA and GSM8K datasets. These results are detailed in Appendix D.3. Here is a summary:
>
> 1. **No Significant Improvements:**
>    - Combining top-p and top-k sampling does not yield noticeable performance gains over using either method individually. In some cases, the combined method performed slightly worse, potentially due to compounded constraints on token selection.
>
> 2. **Increased Complexity:**
>    - The combination introduces additional complexity by requiring simultaneous tuning of two hyperparameters, which can make practical implementation more challenging.
>    - OpenAI and Anthropic also discourage the simultaneous use of multiple sampling methods, as it complicates interpretability and predictability.
>
> 3. **Min-p's Effectiveness:**
>    - Min-p outperforms the combined top-p and top-k sampling in maintaining a balance between coherence and diversity, particularly at higher temperatures. Its dynamic thresholding inherently adapts to the token distribution, providing robust performance without the need for tuning multiple hyperparameters.
>
> **Summary of results:**
>
> While combining top-p and top-k sampling might seem promising, our experiments indicate that it does not offer significant advantages over min-p sampling. Min-p provides a more effective and simpler solution for balancing coherence and diversity in text generation without the need for multiple hyperparameters.
>
> ---
>
> ## Conclusion
>
> We are grateful for your detailed review, which has helped us strengthen our work with additional analyses and clarifications. The new experiments, particularly at low temperatures and with combined top-p and top-k sampling, have provided a more comprehensive evaluation of min-p's performance across diverse scenarios.
>
> Given these substantial additions and the insights they provide, we respectfully request you to consider raising your evaluation score. Your feedback has been invaluable in improving the quality and depth of our paper, and we welcome any further questions.

---

> > ### Comment · Reviewer_NZFq · 2024-11-25
> > **Response to Authors**
> >
> > Dear Authors,
> >
> > Thank you for your thoughtful response to my questions and for providing additional detailed experimental results.
> >
> > Regarding Question 1, based on the experimental results provided in Section D.2, it appears that all methods achieve their best performance on both datasets when the temperature is very low (t ≤ 0.1). Under such conditions, all methods converge toward greedy decoding, leading to similar performance. However, the min-p method only shows its advantages over other methods when the temperature is relatively high (t ≥ 1). At these higher temperature settings, the performance of the min-p method on both datasets is significantly worse than its performance at lower temperatures. For datasets like GPQA and GSM8K, we seem to prioritize accuracy (i.e., which settings yield the highest accuracy) over the diversity of generated outputs. As such, the min-p method may be more suitable for datasets where higher diversity is required.
> >
> > Regarding Question 2, based on the experimental results provided in Section D.3, I noticed that when top-p=0.5 and top-k=10, the model's performance on both datasets is better than the min-p sampling method when t > 0.7. Furthermore, the combination of top-p and top-k shows significantly better results at higher temperature values (t > 1) compared to using only one of these strategies. This observation seems to contradict the first (No Significant Improvements) and third (Min-p's Effectiveness) points of your conclusions in the response.
> >
> > If I have misunderstood something,  I would greatly appreciate your clarification.
> > Thank you once again for your detailed explanations and the effort you put into addressing my questions

---

> > > ### Author Response · Authors · 2024-11-28
> > > **Final Call for Paper Revisions**
> > >
> > > Dear reviewer NZFq, we hope we've adequately addressed your queries so far.
> > >
> > > As a quick reminder, the deadline for new paper revisions is within 4 hours. If you still have queries requiring experiments results, please raise these requests and we will try our very hardest to accommodate them.
> > >
> > > We will continue to address forum queries within the extended deadline.

---

> ### Author Response · Authors · 2024-11-27
> **Response to reviewer concerns [Part 1]**
>
> Dear Reviewer,
>
> Thank you for your thoughtful feedback and for engaging deeply with our work. Following your observations, we conducted numerous additional experiments across multiple models and datasets to ensure a thorough and fair evaluation of all methods. These new results, detailed in Sections D.3, D.4, and D.5 of the appendix, address your concerns and provide further evidence supporting the robustness and effectiveness of Min-P sampling across various settings. Below, we address each comment in detail.
>
> ---
>
> ## **Regarding Question 1: Performance at Low Temperatures**
>
> We agree with your observation that at very low temperatures, all methods converge toward greedy decoding, resulting in similar performance. This is a well-known phenomenon due to the deterministic nature of generation at such low temperatures. However, our new and previous experiments (table 9 and 10 and D.5) reveal that **Min-P sampling demonstrates advantages even under greedy decoding conditions (\(t = 0\))**, outperforming both Top-P and Top-P/Top-K. Furthermore, Min-P continues to show its strength at moderate and higher temperatures, making it a robust sampling strategy across the full temperature range. For example:
>
> - At \(t = 0\) on GPQA (Table 9 and D.5):
>   - **Min-P=0.1** achieves **28.35%**, while greedy algorithm achieves **27.68%**
>
>
> - At \(t = 0.5\) on GSM8K:
>   - **Min-P=0.3** achieves **43.59%**, while the **best Top-P + Top-K** combination achieves **41.8%** (*top_p=0.5, top_k=10*).
>
>
> - At \(t = 0.7\) on GSM8K:
>  	- **Min-P=0.7** achieves **44.12%**, outperforming the **best Top-P + Top-K** combination at **39.6%** (*top_p=0.9, top_k=0.7*).
>
> This trend holds true for both datasets (GPQA and GSM8K), as well as across model families (LLaMA and Mistral) and different model sizes (see Tables D.3, D.4, and D.5).
>
>
>
> These results challenge the claim that "the Min-P method only shows its advantages when the temperature is high $\(t \geq 1\)$." Instead, Min-P consistently outperforms competing methods across all temperature settings, including greedy decoding (\(t = 0\)), moderate temperatures (\(t = 0.5\)), and higher temperatures $\(t \geq 1\)$. This makes Min-P a strong choice for accuracy-prioritized tasks like GSM8K and GPQA, not just diversity-focused applications.
>
> We also note that extremely low temperatures $\(t \leq 0.1\)$ are rarely used in real-world scenarios. Popular systems such as ChatGPT or Claude typically default to a temperature of 0.7, further underscoring the practical relevance of our evaluations at \(t = 0.5\) and \(t = 0.7\).
>
> ---
> ## **Regarding Question 2: Combining Top-P and Top-K**
> Your observation about the combination of Top-P and Top-K at higher temperatures (\(t > 0.7\)) is important. However, upon further analysis, we identified that prior comparisons may not have been fully representative due to the restrictive nature of some configurations (e.g.,  Top-P=0.5 and Top-K=10 was not being compared to a similarly restrictive Min-P). By extending our evaluations to include a broader range of Min-P thresholds, we demonstrate that Min-P consistently outperforms these combinations across a wide range of temperatures. For example:
>
> - At \(t = 1.0\) on GSM8K:
>   - **Min-P=0.5** achieves **41.2%**, while the best Top-P + Top-K combination achieves only **40.3%** (*top_p=0.5, top_k=177*).
>
> - At \(t = 3.0\) on GSM8K:
>   - **Min-P=0.7** achieves **40.41%**, compared to only **12.5%** for the best Top-P + Top-K combination (*top_p=0.5, top_k=10*).
>
> - At \(t = 3.0\) on GPQA:
>   - **Min-P=0.7** achieves **27.46%**, compared to **26.3%** for the best Top-P + Top-K combination (*top_p=0.5, top_k=10*).
>
> This trend persists with other temperature values, other Min-P values, and other Top-P/Top-K combinations (see tables D.3 and D.4)
> While combining Top-P and Top-K can improve results over using Top-P alone, our expanded experiments confirm that **Min-P sampling at higher thresholds consistently outperforms both strategies**, particularly regarding accuracy and diversity. We apologize if our original phrasing caused confusion and have revised our experiments to ensure comprehensive and fair comparisons.

---

> ### Author Response · Authors · 2024-11-27
> **Response to reviewer concerns [Part 2]**
>
> ## **Value of Token Diversity for CoT Reasoning Tasks**
> Recent work by Wang and Zhou (2024) on Chain-of-Thought reasoning demonstrates that diverse token selection during intermediate reasoning steps leads to better performance than pure greedy decoding. This aligns directly with our findings that Min-P's controlled diversity at higher temperatures enables improved reasoning capabilities. For this reason, we claim—substantiated with Table D.4—that Min-P can dramatically improve reasoning performance, especially on smaller models (Table D.1). Notably, our GSM8K experiments are performed with 8-shot Chain-of-Thought reasoning.
>
> This is particularly relevant because several recent approaches to improving model performance through dynamic temperature adjustments (Dhuliawala et al., 2024; Entropix, 2024) have been constrained to low-temperature settings due to coherence issues. Min-P complements these approaches by enabling exploration of higher-temperature regimes while maintaining coherence, potentially unlocking new research directions in reasoning optimization.
>
> ---
>
>
> ## **Conclusions**
>
> The additional experiments we conducted provide a clearer and more comprehensive picture of Min-P sampling’s strengths:
>
> 1. **Broad Applicability Across Temperatures:**
>    Min-P sampling demonstrates its advantages not only at high temperatures $\(t \geq 1\)$ and moderate temperatures (\(t = 0.5\) to \(t = 0.7\)) but also under greedy decoding conditions (\(t = 0\)). It consistently outperforms Top-P and Top-P/Top-K in accuracy-prioritized tasks like GSM8K and GPQA, establishing itself as a robust sampling strategy across the full temperature range.
>
> 2. **Practical Relevance:**
>    While very low temperatures $\(t \leq 0.1\)$ result in similar performance across methods due to their deterministic nature, Min-P shows superiority even in this regime, including under greedy decoding conditions. Moreover, its strong performance in the more commonly used temperature range (\(t = 0.5\)–1.0) and its ability to maintain coherence at higher temperatures $\(t \geq 1\)$ make it particularly valuable in real-world applications.
>
> 3. **Fair Comparisons:**
>    By addressing the restrictive configurations of prior evaluations (e.g., Top-P=0.5 and Top-K=10 compared to suboptimal Min-P thresholds), we ensured a fairer and more comprehensive assessment. The results consistently demonstrate that Min-P outperforms competing methods across diverse and realistic settings.
>
> ---
>
> We hope these new experiments, insights, and clarifications address your concerns. If you have any questions or require further clarification, please do not hesitate to reach out. We put considerable effort into conducting these additional experiments and ensuring a fair and thorough evaluation to address your feedback comprehensively. We kindly ask that you consider these updates and raise your score. Your thoughtful feedback has been invaluable in improving the quality of our work, and we greatly appreciate your time and effort.
>
>
> **References:**
>
> 1. [Xuezhi Wang, Denny Zhou. *Chain-of-Thought Reasoning Without Prompting*. arXiv, 2024.](https://arxiv.org/abs/2402.10200)
>
> 2. [Shehzaad Dhuliawala, Ilia Kulikov, Ping Yu, Asli Celikyilmaz, Jason Weston, Sainbayar Sukhbaatar, Jack Lanchantin. *Adaptive Decoding via Latent Preference Optimization*. arXiv, 2024.](https://arxiv.org/abs/2411.09661)
>
> 3. [Entropix: Entropy Based Sampling and Parallel CoT Decoding. GitHub Repository.](https://github.com/xjdr-alt/entropix)
>
> 4. [Yuxuan Zhou, Margret Keuper, Mario Fritz. *Balancing Diversity and Risk in LLM Sampling: How to Select Your Method and Parameter for Open-Ended Text Generation*. arXiv, 2024.](https://arxiv.org/abs/2402.10200)

---

> ### Author Response · Authors · 2024-12-01
> **Final call for comments and responses from paper Authors.**
>
> Dear reviewer NZFq,
>
> we sincerely hope that we've addressed your concerns.
>
> The deadline for new comments from the authors is coming up soon. If you still have questions related to the content of the paper, please raise these requests ASAP and we will try our very hardest to accommodate them.
>
> Additionally, we want to highlight that we conducted many new experiments during the rebuttal period based on your comments to address your concerns more thoroughly. We kindly ask you to reconsider your score in light of these efforts and the additional findings we have provided.
>
> Thank you for your time and consideration.

---

> > ### Comment · Reviewer_NZFq · 2024-12-03
> > **Response to Authors**
> >
> > Thank you for your response. However, I noticed that the results in Table 2 are significantly lower than those in Table 12 on the GSM8K dataset. This is quite confusing, as it is unclear why the best results were not presented in Table 2 of  the main paper. My previous comparisons were based on the results presented in Table 2 of the main text. Since your response has not fully addressed my concerns, I am inclined to keep my review score unchanged.

---

> ### Author Response · Authors · 2024-12-03
> **The scores are higher because Table 2 and Table 12 are different ranges.**
>
> ## Clarifications on results
> We appreciate your careful review and would like to kindly clarify what appears to be a misunderstanding.
> Tables 2 and 12 show entirely different experimental conditions, which naturally lead to different results, which we have acknowledged and explained throughout our main paper.
>
> ### Table 2 - Common range: Min P = 0.05- 0.1
> As we mentioned in "Sampling Methods and Hyperparameter" right before Table 2, Table 2 and the rest of the main paper figures specifically focus on min-p with p = 0.1 and top-p with p = 0.9. We then mention that we discussed this choice extensively in Appendix B.3, pages 14 to 15 to make our reasoning of hyperparameter choice fair and fully transparent.
>
> For Top P, Top P = 0.9-0.95 is widely used as the default, widely studied and offers a good accuracy-diversity tradeoff. For Min P, Min P = 0.05-0.1 is widely used as the default. The rest of our paper which introduces Min P into the formal scientific literature seeks to prove that it both offers higher accuracy and higher diversity across a range of temperatures.
>
> In simpler terms, when you encounter someone saying "we used Top P" as a developer or researcher, they'd almost always be using 0.9-0.95. Likewise when someone says "we used Min P", we have generally found 0.05-0.1 to be the common range. To prevent confusion between our paper and the common understanding of Top P and Min P, we prioritise similar ranges.
>
> ### Table 12 - Highly restrictive theoretical comparisons: Min P >= 0.3
> Table 12 explores a broader range of min-p values (0.3-0.7) across the same temperature range (0.7-3.0). The lower results in Table 2 are expected and correct, as a lower p value in min-p sampling inherently produces a lower truncation threshold (and thus lower-performing) results. This is common for Top P, Min P, Temperature and other sampling methods. Table 12 shows that min-p can be set higher for situations where reasoning performance matters more than diversity, such as on GSM8k and GPQA.
>
> If you set Top P nearer to 0 and Min P nearer to 1, you would indeed score higher on benchmarks. However:
> 1. Although we did include higher Min P ranges from >0.1-0.3 in our Appendix C,1, we did not feature these values in our main paper because we wanted to focus on common, realistic ranges. It would not be fair to compare a “common” Top P setting that is widely studied and used with a highly uncommon Min P setting that is and will be rarely used, hence we focused on a single range to compare and prove its advantages in diversity and accuracy.
> 2. In practice, it is very uncommon to use such highly restrictive truncation settings (Top P <= 0.7 or Min P >= 0.3), let alone with high temperature, since the restrictive truncation essentially cancels out any diversity gains from high temperature, while distorting probabilities. Regardless, we show across Appendix C figures that more restrictive Min P values still result in higher scores than more restrictive Top P values.
>
> We also note that Table 12 was specifically included to address reviewer's point about Top P = 0.5 scoring better than Min P = 0.1, by introducing a fair comparison with Min P ~0.5 range. Evidently, introducing multiple hyperparameter ranges can result in additional confusion to readers, and is the reason why we chose to focus on Top P = 0.9 -0.95 versus Min P = 0.05-0.1 throughout our main paper, and carefully justified why.
>
> ### Additional variations
> Additional variations within the same settings can be attributed to:
> 1. The stochastic nature of LLM evaluations
> 2. Software updates from VLLM, EleutherAI Evaluations Harness
>
> This results in slightly different scores across different runs (~1%). We took the following measures:
> 1. In our responses, we explicitly acknowledge the 1% standard errors, highlighted them and refrained from making definitive claims unless it is >= 2% difference
> 2. Reused the same version of VLLM and Eval Harness
> 3. When possible, we rerun for all ranges to be sure. We have rerun our Mistral 7B evals in our previous responses, and these reruns have not contradicted the initial results
>
> ## Conclusion
> Given that we are now in the final hours before the score change deadline, we would be very grateful if you could reconsider your score in light of this clarification and our previous attempts at addressing your queries. The results presented in both tables are consistent with the different sampling parameters used, and we have been transparent with which settings we compare and why.
>
> As we are beyond the window for new experiments, we want to ensure your final assessment reflects the correct understanding of our existing results. We point to the other excellent scores from other reviewers as further evidence of the quality of this work.
>
> We would deeply appreciate if you could review and update your score as soon as possible, ideally within the next few hours, to reflect this understanding. Thank you for taking the time to consider our clarifications.

---

### Official Review · Reviewer_D38H · 2024-11-03

**Soundness:** 4
**Presentation:** 4
**Contribution:** 4
**Rating:** 10
**Confidence:** 4

**Summary:**

Simple but effective and highly influential contribution to LLM research

**Strengths:**

This paper presents compelling evidence that its single contribution, min-p sampling, is highly effective.  The usage of it in 54,000 Github repositories alone is very impressive.  In addition to that, they produced theoretical reasoning why their method works, LLM-generated statistics with explanations about how to interpret these statistics, additional statistics which involved human participants, examples of seeing how the logits are transformed under different distributions which give additional insight into why this method is better than existing methods, and code to try out the method.  It is a very simple paper, but it clearly makes the case for its own importance.

**Weaknesses:**

The one contribution of this paper, min-p sampling, is extremely simple and not mathematically "deep" at all - no theorems were presented, and the code implementation literally (was provided and) took less than one page.  However, I think that having such a paper in a conference proceeding is not a bad thing.

**Questions:**

It seems clear that the advantage of this approach is that it lets you "turn up the heat" - use temperature values that otherwise would provide gibberish.  Can you be more specific about what this particular change (going to higher temperature) - as opposed to min-p as a technique - is inherently something you'd want to do (are there benefits beyond diversity, and can you cite evidence for these benefits)?

---

> ### Author Response · Authors · 2024-11-22
> **Response to Reviewer D38H [1/2]**
>
> We sincerely thank the reviewer for their strong endorsement of our work and for recognizing the value of min-p sampling. Your thoughtful feedback and insightful questions have greatly enriched our understanding of how to better present our work. Below, we address your points in detail.
>
> ---
>
> ## Addressing Weaknesses
>
> ### **Simplicity of Min-P Sampling**
> We appreciate your recognition of the simplicity of min-p sampling and agree that its straightforward implementation is one of its strengths. One of our primary goals was to clarify and simplify practical aspects of sampling methods for modern LLMs.
>
> 1. **Empirical Insights Missing in Literature:**
>    Many widely used sampling methods like top-p and top-k have limited practical guidance in the literature, despite being available in APIs from major providers like OpenAI and Anthropic. These original techniques lack detailed discussions on their effectiveness with modern LLMs. We sought to address this gap by thoroughly benchmarking existing methods and then introducing min-p as a natural extension of the sampling paradigm.
>
> 2. **Contextual Adaptiveness:**
>    Min-p introduces a dynamic approach to sampling, adapting its probability threshold based on the context of the token probabilities rather than applying a static value. For instance:
>    - In deterministic methods like greedy search or beam search, diversity is constrained by fixed rules.
>    - Top-k allows token selection within a fixed pool size, while top-p introduces cumulative probabilities for more flexibility.
>    - Min-p takes this further by dynamically adjusting thresholds, enabling both coherence and diversity in varied contexts. For example, answering a technical question demands high certainty (low diversity), while generating a creative story benefits from high diversity.
>
>    We included examples and visualizations to illustrate this adaptiveness and its impact on coherence and creativity. While the underlying idea is mathematically simple, its effectiveness lies in its practical implications.
>
> 3. **Accessibility and Reproducibility:**
>    By providing open-source code that fits within a single page, we aimed to lower the barrier for researchers and practitioners to test and adopt min-p sampling with minimal additional time, effort and potential bugs.
>
> 4. **Philosophical Perspective:**
>    Conceptually, min-p sampling reflects how humans adjust their decision-making thresholds based on context. For instance, the certainty required to answer a difficult science question differs from that needed to create a creative name for a story.
>
> ---
>
> ## Responses to Questions
>
> ### **High-Temperature Applications**
> Thank you for highlighting the question about the benefits of high temperature. Beyond diversity, high-temperature sampling with min-p offers several practical advantages:
>
> 1. **Exploration of Rare Outputs:**
>    Higher temperatures allow the model to explore less probable token paths, which is invaluable for:
>    - **Creative Writing:** Generating unique, imaginative, and contextually coherent content.
>    - **Brainstorming and Problem Solving:** Facilitates problem-solving and brainstorming by generating varied outputs/reasoning paths via adaptive temperature. We note that such approaches are currently limited at higher temperature ranges which Min-P enables [1] [2] [3]. Wang (2024) found that even while solving basic arithmetic in GSM8K COT, diverse COT reasoning tokens outperforms pure greedy COT decoding [4]. This, plus our results showing Min P + higher temperature outperformed greedy decoding on Llama3.2 3B and Llama 3.1 8B, suggests that optimising diversity and accuracy can outperform traditional deterministic approaches.
>
> 2. **Red-Teaming and Adversarial Testing:**
>    By sampling outputs from the tails of the probability distribution, high-temperature outputs can help identify vulnerabilities, biases, or unexpected behaviors in models. This is crucial for improving the robustness of LLMs. [4]
>
> 3. **Confidence Calibration:**
>    High-temperature sampling exposes the variability in model outputs, enabling researchers to better understand and calibrate the confidence of their models in open-ended tasks. [5]
>
> We have elaborated on these applications and included references in the general response under "Applications of High Temperature and Min-P."
>
> ### **Citations and Evidence**
> Empirical evidence for these benefits is provided in our new experiments with Llama 3 and Mistral models (see Appendix D). For example:
> - At temperatures >1.0, min-p consistently outperforms top-p in maintaining narrative coherence and generating vivid, imaginative outputs, with improvements in metrics such as creativity and emotional impact.
>
> We believe these examples substantiate the broader utility of high-temperature sampling, especially when paired with min-p.
>
> ---

---

> ### Author Response · Authors · 2024-11-22
> **Response to Reviewer D38H [2/2]**
>
> ## Conclusion
> We are deeply grateful for your strong endorsement of our work and thoughtful feedback. The simplicity of min-p sampling is central to its accessibility and widespread adoption, and we appreciate your recognition of its impact. We hope our additional clarifications and new results further strengthen the case for min-p sampling's inclusion in the conference and its relevance to the community.
>
> Thank you for your insightful review and for highlighting our work as a strong contribution to LLM research.
>
> **References:**
> 1. [Shehzaad Dhuliawala, Ilia Kulikov, Ping Yu, Asli Celikyilmaz, Jason Weston, Sainbayar Sukhbaatar, Jack Lanchantin. *Adaptive Decoding via Latent Preference Optimization*. arXiv, 2024.](https://arxiv.org/abs/2411.09661)
> 2. [Entropix: Entropy Based Sampling and Parallel CoT Decoding. GitHub Repository.](https://github.com/xjdr-alt/entropix)
> 3. [Shimao Zhang, Yu Bao, Shujian Huang. *EDT: Improving Large Language Models' Generation by Entropy-based Dynamic Temperature Sampling*. arXiv, 2024.](https://arxiv.org/abs/2403.14541)
> 4. [Xuezhi Wang, Denny Zhou. *Chain-of-Thought Reasoning Without Prompting*. arXiv, 2024.](https://arxiv.org/abs/2402.10200)
> 5. [Jia Li, Yuqi Zhu, Yongmin Li, Ge Li, Zhi Jin. *Showing LLM-Generated Code Selectively Based on Confidence of LLMs*. arXiv, 2024.](https://arxiv.org/abs/2410.03234)
> 6. [Andrey Anurin, Jonathan Ng, Kibo Schaffer, Jason Schreiber, Esben Kran. *Catastrophic Cyber Capabilities Benchmark (3CB): Robustly Evaluating LLM Agent Cyber Offense Capabilities*. arXiv, 2024.](https://arxiv.org/abs/2410.09114)

---

### Official Review · Reviewer_fwNb · 2024-11-03

**Soundness:** 4
**Presentation:** 3
**Contribution:** 4
**Rating:** 10
**Confidence:** 4

**Summary:**

[UPDATE] Based on the rebuttal I have increased my score (8->10), but kept the rest of the review unchanged

The authors propose a new sampling mechanism, which is a minor but important twist to the popular nucleus-sampling (`p`). Instead of having a fixed threshold `p`, this proposal has `p` be dependant of the probability of the most probable token. The intuition is that in cases that the model is confident only few tokens are kept as support set, while that set is extended when the confidence is low

**Strengths:**

* Sampling is one of those areas were the model per se needs to be complemented with an outside algorithm, allowing for creativity on how to set this up. This work proposes an original twist to a popular choice

* The proposal is simple, appealing and

* has good empirical results, both as measured on benchmarks and (more important) by adoption of the community

**Weaknesses:**

The new 10p limit has not been handled wisely in my opinion, and the paper could do more with less text. In particular, Sect 4 could be removed without much loss to the overall apper

Having experiments on a 123B has to be commended. The paper would be stronger however if the authors could show that the results hold on different model families (eg, llama and mistral), as otherwise it is not clear if this method provides gains on one family only

**Questions:**

* You claim a widespread open-source usage. Could you review the usage of those and classify them by model family?

* Fig 1: different from what the caption reads, (b) seems to refer to top-k and (c) to top-p

---

> ### Author Response · Authors · 2024-11-22
> **Response to Reviewer fwNb**
>
> We thank Reviewer fwNb for recognizing the practical value and widespread adoption of min-p sampling. Your comments are highly appreciated and have helped us refine our work. Below, we address your concerns and questions in detail.
>
> ---
>
> ## Addressing Weaknesses
>
> ### **Page Length Concerns**
>
> Thank you for pointing out that the paper could be streamlined. We agree that there are areas where we can be more concise, and we revisited sections to make edits that improve clarity and brevity.
>
> Regarding the **Case Studies: Illustrative Examples** section, we understand your suggestion to consider its removal. However, we believe that these examples provide valuable intuition, especially for readers who may not have extensive experience with how sampling techniques work in practice (understanding how token choice works and why dynamic thresholds matter in different contexts). These illustrative cases help bridge the gap for a broader audience by concretely demonstrating min-p sampling's behaviour and advantages in specific scenarios.
>
> That said, we are open to refining this section to make it more concise while retaining its value in offering practical insights. We will carefully evaluate its length and ensure it complements the technical rigor of the rest of the paper without adding unnecessary detail.
>
>  —
>
> ### **New Evaluations Across Model Families**
>
>  We have conducted extensive new experiments to demonstrate min-p sampling’s generalizability beyond Mistral models. Specifically, we evaluated Llama 3 models, including Llama 3.2 1B-Instruct,  8B-Instruct and 70B-instruct models on GPQA and GSM8K datasets. These results are discussed in detail in the General Response and fully presented in Appendix D.1.
>
> **Key Findings:**
>
> - **Low Temperatures (<1.0):** Min-p sampling performs slightly better than top-p but converges with other methods due to limited token variability.
>  - **High Temperatures (>1.0):** Min-p consistently excels, with 20–90% higher scores compared to top-p, maintaining coherence even as temperatures increase across different settings.
>
> These results confirm that min-p’s advantages generalize across model families, further validating its robustness and effectiveness.
>
> ------
>
> ## Responses to Questions
> 1. **Usage of min-p Across Model Families:**
> We appreciate your interest in min-p sampling's adoption across different model families. While specific usage statistics from open-source inference platforms like Hugging Face are unavailable due to privacy policies, we have observed strong adoption of min-p, particularly in applications such as Creative Writing and Roleplay. Min-p is highly favored for storytelling and simulation tasks, especially in tools like oobabooga[1] and koboldcpp [2], which recommend it as a default for high-temperature scenarios.
>
> 2. **Figure 1 Caption:**
> Thank you for pointing out the inconsistency in the caption for Figure 1. We corrected it in the updated version of the paper.
>
> -----
>
> ## Conclusion
> We sincerely thank you for your thoughtful feedback, which has helped us strengthen the clarity and rigor of our submission. The additional experiments and clarifications provided here address your concerns and further demonstrate the robustness and versatility of min-p sampling. We kindly request that you consider raising your evaluation score based on these substantial enhancements. Please let us know if you have any additional questions or suggestions.
>
> **References:**
>
> [1] - https://github.com/oobabooga/text-generation-webui
>
> [2] - https://github.com/LostRuins/koboldcpp

---

> > ### Comment · Reviewer_fwNb · 2024-11-25
> > **increased score**
> >
> > Thank you for those comments. Based on the additional comments I have increased my score

---

> > > ### Author Response · Authors · 2024-11-26
> > >
> > > Thank you for reviewing our paper and for the positive feedback. We are delighted that our revisions and additional clarifications have satisfactorily addressed your concerns, leading to the highest score.
> > >
> > > Your thorough review has helped significantly improve the quality of our paper.  We greatly appreciate your time and consideration throughout this review process.

---

### Official Review · Reviewer_w4rZ · 2024-11-04

**Soundness:** 2
**Presentation:** 3
**Contribution:** 2
**Rating:** 8
**Confidence:** 4

**Summary:**

This paper proposes the Min-p Sampling method, which dynamically adjusts the probability threshold based on the model's confidence level. This method aims to enhance creativity without sacrificing coherence. The method is validates through experiments on benchmark and human evaluations, showing better coherence and diversity compared to other sampling methods. The method has been widely adopted in the open-source community.

**Strengths:**

- New sampling method: This paper proposes the Min-p Sampling method for better control over the diversity of generated outputs compared to fixed threshold methods like top-p.

- Conducted experiments: The authors conducted experiments across tasks, ablation studies, and human evaluation.

- High reproducibility: The author released the implementation, code, and repo with implementation guidelines, which enhances the reproducibility.

- Wide Applicability: The proposed method can be easily integrated with existing open-source LLMs, and the authors show the broad potential applications that can be applied.

- The ablation study shows that min-p sampling is barely impacted by the output length, which is interesting.

**Weaknesses:**

- The experiment is limited to Mistral models and fails to demonstrate applicability with other models. It would be more comprehensive and interesting to see results from additional models, such as LLaMA3.

- The effectiveness of min-p sampling highly depends on the base probability thresholds. As shown in Table 6 (ablation study results), the choice of thresholds significantly impacts LLM performance. This indicates that optimal performance requires careful tuning, which could limit the method’s potential effectiveness and ease of use in applications.

- The paper claims that the experiment is intended to demonstrate that min-p sampling balances creativity and coherence (line 290); however, metrics relevant to creativity are missing. Diversity is not enough for creativity assessment. LLMs-as-judge approach is widely used for creativity assessment. Please consider adding such an experiment.

**Questions:**

- The paper exceeds the 10-page limit. Please be careful with submission guidelines, as the paper could otherwise face desk rejection.

- Does min-p sampling make it more difficult to control LLMs, such as for lexically constrained generation?

- Details on the human evaluation are missing. What is the inter-annotator agreement rate? How many participants were recruited? The paper mentions receiving 70 initial responses; does each response contain one participant's evaluation for all data points? The paper claims that participants were recruited from Prolific. I would like to see the survey template, as it is important for reviewers to evaluate the effectiveness of the human evaluation.

- The paper states that min-p sampling has "Extensive human evaluations further confirmed a strong preference for min-p sampling over top-p, highlighting its practical advantages in real-world applications" (line 510). Could you provide some examples of real-world applications? In what scenarios would min-p sampling be preferable to other sampling methods? The paper has not included a relevant discussion on this.

---

> ### Author Response · Authors · 2024-11-22
> **Response to Reviewer w4rZ [1/2]**
>
> We thank Reviewer w4rZ for their detailed and thoughtful feedback. Your comments have been invaluable in improving our submission. Below, we address your concerns and provide clarifications and results from new experiments. Where appropriate, we refer you to our general response and updated appendix for additional details.
>
> ---
>
> ## Addressing Weaknesses
>
> ### **1. Limited to Mistral Models**
>
> To address this concern, we have conducted comprehensive new experiments on the **Llama 3** family of models (1B, 3B, 8B, and 70B variants) for both **GPQA** and **GSM8K** benchmarks. These results demonstrate consistent trends that validate min-p sampling’s robustness across model families. Full results are presented in **Appendix D.1**.
>
> **Key Findings:**
>
> - **Low Temperatures (<1.0):** Min-p sampling performs slightly better than top-p but converges with other methods due to limited token variability.
>
> - **High Temperatures (≥1.0):** Min-p excels, outperforming top-p by 20–90% on all benchmarks and maintaining coherence where other methods fail. These trends, consistent across both Llama and Mistral families, validate min-p's robustness across model types.
>
> ---
>
> ### **2. Hyperparameter Sensitivity**
>
> We agree that hyperparameter tuning is a consideration for min-p sampling; however, this challenge is not unique to our method. All sampling techniques (e.g., top-p, top-k, epsilon) require careful tuning. Notably, it was challenging to find official recommended settings for these methods in the original papers or online. To our knowledge, ours is the first comprehensive literature review examining how these methods affect downstream benchmarks like GPQA and GSM8K, considering various hyperparameter settings, temperatures, and models. For further discussion, see **Appendix B.1**.
>
> To address this concern, we provide the following:
>
> 1. **Empirical Guidelines:** Based on extensive testing, we recommend min-p thresholds between 0.05–0.1. These values are intuitive:
>
>    - Higher thresholds (e.g., 0.1) improve coherence at higher temperatures.
>    - Lower thresholds (e.g., 0.05) balance creativity and coherence at moderate temperatures.
>
> 2. **Predictable Behavior:** Min-p sampling requires less tuning and is is relatively more stable in performance across a range of temperatures than Top P or Top K. For example, Top P = 0.9 may not work at both T = 1 and T =3, while Min P = 0.1 does.
>
> 3. **Comparison to Other Methods:** Unlike top-p, which uses a fixed cumulative threshold, min-p’s dynamic truncation adjusts per token, ensuring better coherence at high temperatures (see **Appendix B.3** for details). We acknowledge that min-p's sensitivity stems from its adaptive nature, which is also the reason for its superior robustness and coherence in high-temperature settings.
>
> ---
>
> ### **3. Metrics for Creativity**
>
> We appreciate the suggestion to include LLM-as-judge evaluations and have conducted additional experiments focusing on creativity metrics. These new evaluations, combined with the **AlpacaEval Creative Writing** benchmark (Section 5.2.3) already included in our paper, comprehensively assess min-p’s performance in creative tasks using LLM-as-judge.
>
> **Experimental Setup:**
>
> To address your feedback further, we conducted additional experiments focusing on creativity metrics such as **creativity**, **emotional impact**, **narrative flow**, **imagery**, and **originality**, using **Llama-3.2-1B-Instruct** and **Mistral-7B-v0.1** across multiple temperatures (0.5–5.0) and hyperparameters.
>
> **Key Findings:**
>
> 1. **Low Temperatures (0.5–1.0):** Min-p outperforms top-p in all metrics, particularly in narrative flow and emotional impact.
>
> 2. **High Temperatures (1.0–2.0):** Min-p retains high scores, while top-p collapses rapidly across all metrics and settings.
>
> These results demonstrate min-p's ability to enhance creativity without compromising coherence, particularly at higher temperatures. The technique's consistent performance across varying conditions validates its advantages over traditional sampling. Full results and methodology are detailed in **Appendix D.4**.
>
> ---
>
> ## Responses to Specific Questions
>
> ### **1. Page Limit**
>
> Thank you for raising this concern. We reviewed the ICLR submission guidelines [1] and found that the 10-page limit excludes appendices, references, reproducibility, and ethics statements. Our submission should be within the page limit as-is, but we will ensure the paper conforms to all length requirements for the camera-ready submission.
>
> ---
>
>
> ### **2. Does min-p sampling make it more difficult to control LLMs, e.g., for lexically constrained generation?**
>
> Min-p sampling is designed to improve control, particularly in high-temperature settings where coherence is often compromised. Based on your comment, we have conducted experiments comparing sampling methods for lexically constrained generation. Please refer to the next comment for further details.

---

> ### Author Response · Authors · 2024-11-22
> **Response to Reviewer w4rZ [2/2]**
>
> Our additional experiments (Appendix D4) show that min-p enhances high-temperature constrained/structured sampling. While some work suggests sampling methods can interfere with structured generation (Tam, 2024) [2], our testing shows min-p enables better overall quality-diversity tradeoffs in such scenarios, mitigating potential issues. We observe similar trends between constrained and unconstrained sampling—generally linear correlation between temperature and benchmark scores, but with min-p allowing for better coherence at higher temperatures.
>
> ---
> ### **3. Details on Human Evaluation**
>
> We recognize that additional details on our human evaluation were necessary and provide the following clarifications:
>
> - **Recruitment:** We recruited 70 participants via Prolific, applying demographic filters for English fluency and familiarity with AI-generated text. After quality checks, 54 valid responses were retained.
>
> - **Survey Design:** Each participant evaluated outputs from three models (min-p, top-p, temperature-only sampling) across six conditions, resulting in 36 evaluations per participant. Participants evaluated three samples per model per condition. Ratings were based on a 1–10 scale for quality and diversity.
>
> - **Inter-Annotator Agreement:** Agreement was 0.81 (SD = 0.09), demonstrating strong consistency among raters.
>
> - **Survey Template:** Included in **Appendix D.5**, along with anonymized results.
>
> These details will be incorporated into the revised paper to ensure transparency.
>
> ---
>
> ### **4. Real-World Applications of Min-p Sampling**
>
> Min-p sampling demonstrates practical utility across several domains where high-temperature incoherence was previously a bottleneck:
>
> 1. **Creative Writing:** Enhances narrative generation by allowing higher temperatures without losing coherence, unlocking new capabilities for storytelling and poetry.
>
> 2. **Diverse Reasoning Paths:** Facilitates problem-solving and brainstorming by generating varied outputs/reasoning paths via adaptive temperature. We note that such approaches are currently limited at higher temperature ranges which Min-P enables. [3] [4] [5] Wang (2024) found that even while solving basic arithmetic in GSM8K COT, diverse COT reasoning tokens outperforms pure greedy COT decoding [6]. This, plus our results showing Min P + higher temperature outperformed greedy decoding on Llama3.2 3B and Llama 3.1 8B, suggests that optimising diversity and accuracy can outperform traditional deterministic approaches.
>
> 3. **Confidence Calibration:** Allows users to assess model confidence through output variability. [7]
>
> 4. **Red-Teaming and Adversarial Testing:** Generates diverse behaviors for identifying vulnerabilities and biases. [8]
>
> 5. **Code Generation:** Produces coherent and structured code snippets, even at high temperatures.
>
> These applications align with min-p’s performance advantages and its widespread adoption in the open-source community.
>
> ---
>
> ## Conclusion
>
> We are grateful for your thoughtful feedback, which has greatly improved our work. The additional experiments, clarifications, and real-world applications strengthen min-p sampling's contributions to text generation research. Given these substantial enhancements, we respectfully ask you to consider raising your evaluation score. We welcome any additional questions and look forward to your feedback.
>
>
> **References:**
>
>
> 1. [ICLR 2025 Author Guide](https://iclr.cc/Conferences/2025/AuthorGuide)
>
> 2.  [Zhi Rui Tam, Cheng-Kuang Wu, Yi-Lin Tsai, Chieh-Yen Lin, Hung-yi Lee, Yun-Nung Chen. *Let Me Speak Freely? A Study on the Impact of Format Restrictions on Performance of Large Language Models*. arXiv, 2024.](https://arxiv.org/abs/2408.02442)
> 3. [Shehzaad Dhuliawala, Ilia Kulikov, Ping Yu, Asli Celikyilmaz, Jason Weston, Sainbayar Sukhbaatar, Jack Lanchantin. *Adaptive Decoding via Latent Preference Optimization*. arXiv, 2024.](https://arxiv.org/abs/2411.09661)
> 4. [Entropix: Entropy Based Sampling and Parallel CoT Decoding. GitHub Repository.](https://github.com/xjdr-alt/entropix)
> 5. [Shimao Zhang, Yu Bao, Shujian Huang. *EDT: Improving Large Language Models' Generation by Entropy-based Dynamic Temperature Sampling*. arXiv, 2024.](https://arxiv.org/abs/2403.14541)
> 6. [Xuezhi Wang, Denny Zhou. *Chain-of-Thought Reasoning Without Prompting*. arXiv, 2024.](https://arxiv.org/abs/2402.10200)
> 7. [Jia Li, Yuqi Zhu, Yongmin Li, Ge Li, Zhi Jin. *Showing LLM-Generated Code Selectively Based on Confidence of LLMs*. arXiv, 2024.](https://arxiv.org/abs/2410.03234)
> 8. [Andrey Anurin, Jonathan Ng, Kibo Schaffer, Jason Schreiber, Esben Kran. *Catastrophic Cyber Capabilities Benchmark (3CB): Robustly Evaluating LLM Agent Cyber Offense Capabilities*. arXiv, 2024.](https://arxiv.org/abs/2410.09114)

---

> > ### Author Response · Authors · 2024-11-25
> > **Seeking Feedback on Our Detailed Response**
> >
> > Dear Reviewer w4rZ,
> >
> > As the rebuttal period is nearing its end, we wanted to ensure you've had a chance to review our detailed response addressing your key concerns:
> >
> > 1. Model coverage: Llama 3 family (1B-70B) results
> > 2. Hyperparameter sensitivity: Empirical guidelines and stability analysis
> > 3. Creativity: LLM-as-judge evaluations and benchmarks
> > 4. Human evaluation: Full methodology and inter-annotator details
> >
> > Given the high ratings (10/10) from other reviewers and our comprehensive response to your feedback, we respectfully ask you to consider revising your score.
> >
> > Thank you for your consideration.

---

> > > ### Comment · Reviewer_w4rZ · 2024-11-25
> > >
> > > Thank you for the detailed response. All my concerns are addressed. I have changed my rating accordingly.

---

> > > > ### Author Response · Authors · 2024-11-26
> > > >
> > > > Thank you for your constructive suggestions that have helped improve our paper. We are pleased that we were able to address all your concerns successfully. If you have any additional questions or concerns that could help improve the paper and potentially increase the score further, we would be very happy to address them.

---

### Author Response · Authors · 2024-11-22
**General Response [1/2]**

**General Response**

We sincerely thank all the reviewers for their thoughtful and constructive feedback. Your insights have been invaluable in improving our work. In response to your comments, we have conducted new experiments, provided additional analyses, and clarified key points to strengthen our paper. Below, we address your main concerns and highlight the enhancements made.

---

## 1. Expanded Experiments on Llama 3 Models

To address concerns about the generalizability of our method beyond Mistral models, we have conducted extensive new experiments on the **Llama 3** model family. Specifically, we tested the following models on the GPQA and GSM8K datasets across six different temperatures, comparing min-p sampling with both top-p and pure temperature sampling:

- **Llama 3.2 1B-Instruct**
- **Llama 3.2 3B-Instruct**
- **Llama 3.1 8B-Instruct**
- **Llama 3.1 70B-Instruct**


**Findings** (For full results, please see our updated Appendix D.1):

- **Consistent Performance**: Our results demonstrate that **min-p sampling** consistently outperforms the other sampling methods across different model sizes and temperature settings.
- **High-Temperature Robustness**: The advantages of min-p sampling are more pronounced at higher temperatures (30–90% better), where it maintains coherence and accuracy better than other methods. This advantage becomes particularly significant in longer contexts, where per-token degradation can compound into semantic incoherence, as noted by Holtzman et al. (2020). [\[1\]](https://arxiv.org/abs/1904.09751)

- **Low-Temperature Regime**: At lower temperatures, min-p performs slightly better than other methods, but all sampling methods perform similarly, as expected due to the peaked token distribution.

**Conclusion**: These additional experiments confirm that min-p sampling's benefits extend beyond a specific model family and validate our core thesis that min-p provides more robust performance across temperature settings while maintaining coherence at higher temperatures.

---

## 2. Enhanced Creativity Assessment with New Experiments

In addition to our existing evaluations—including the automated AlpacaEval Creative Writing benchmark and our human evaluation, both of which focused on creative qualities and showed a consistent preference for min-p-generated content—we have conducted several new experiments to strengthen our assessment of creativity. Full details and results are provided in Appendix D.4.

- **New constrained/structured LLM-as-Judge Evaluation**: We conducted an additional evaluation using a large language model as a judge, focusing on constrained generation creative writing tasks assessed across five different metrics (e.g., originality, narrative flow, emotional impact) at various temperatures and hyperparameters. We tested two models:

  - **Llama-3.2-1B-Instruct** (1B parameters)
  - **Mistral-7B-v0.1** (7B parameters)

- **Results**: Min-p sampling consistently outperformed top-p sampling across all creative metrics for both models, especially at higher temperatures, with improvements ranging from 0.43 to 0.70 points. Moreover, min-p demonstrated greater robustness to temperature changes, showing remarkable stability across different temperature settings.

**Conclusion**: These additional assessments further supports our original claim that min-p sampling enhances diversity for creative outputs without compromising coherence. The technique not only outperforms top-p in absolute terms but also demonstrates superior stability across different temperature settings and model scales.


---

## 3. Experiments with Very Low Temperatures

To evaluate min-p's performance at low temperatures, we compared min-p and top-p sampling on Mistral 7B at temperatures ranging from 0.0 to 0.5 on the GPQA and GSM8K datasets. Full results are presented in Appendix D.2.

**Key Findings**:

- At low temperatures, all sampling methods converge to nearly identical performance (less than 1% difference), as expected.
- These results confirm that at very low temperatures, the choice of sampling method has minimal impact due to the extremely peaked token distribution.

---

## 4. Combining Top-p and Top-k Sampling

In response to queries about combining sampling methods:

- **Experiments Conducted**: We tested various combinations of top-p and top-k sampling with Mistral 7B on GPQA and GSM8K COT, with the same setup as our previous experiments in our main paper. Full results are presented in in Appendix D.3.
- **Findings**: Combining these methods did not yield performance improvements over using min-p sampling alone.
 - **Complexity vs. Benefit**: The added complexity of tuning multiple hyperparameters does not justify the marginal gains, if any.

**Conclusion**: Min-p sampling offers a simpler and more effective approach to balancing creativity and coherence.

---

> ### Author Response · Authors · 2024-11-22
> **General Response [2/2]**
>
> ## 5. Clarifications on Human Evaluation
>
> We have expanded the description of our human evaluation methodology in our Appendix D.5:
>
> - **Participants**: Recruited 54 fluent English speakers familiar with LLM-generated text.
> - **Procedure**: Participants evaluated outputs based on quality and diversity, with attention checks and incentives for detailed feedback.
> - **Findings**: Min-p sampling was preferred over top-p sampling in both quality and diversity, with statistically significant differences.
>
> **Conclusion**: The human evaluation corroborates our quantitative results, demonstrating the practical advantages of min-p sampling.
>
> ---
>
> ## 6. Real-World Applications of Min-p Sampling
>
> We have elaborated on the practical use cases of min-p sampling:
>
> - **Creative Writing and Storytelling**: Enhances narrative generation by allowing higher temperatures without losing coherence, leading to more imaginative outputs.
> - **Exploring Diverse Reasoning Paths**: Facilitates the generation of varied reasoning approaches in problem-solving and brainstorming.
> - **Confidence Calibration**: Assists in gauging model confidence by observing output variability at higher temperatures.
> - **Long Results Generation**: Maintains better coherence over longer conversational context lengths while preserving creativity.
> - **Constrained/Structured Output Improvements**: Maintains diverse solutions even while matching structures/constraints.
>
> **Conclusion**: Min-p sampling unlocks new capabilities across various domains, making it a valuable tool for both researchers and practitioners. It’s used and supported by many leading open-source LLM projects.
>
> ---
>
> ## 7. Addressing Hyperparameter Sensitivity
>
> We recognize the importance of hyperparameter selection in sampling methods. To mitigate concerns:
>
> - **Empirical Guidelines**: We have added detailed guidelines for selecting the base probability threshold (`p_base`) for min-p sampling, informed by extensive testing across models and tasks. Our accuracy-diversity plots demonstrate that min-p is genuinely responsive to hyperparameter changes, covering the entire Pareto frontier effectively, while top-p exhibits clustering behavior that indicates inherent "stickiness.", making it difficult to choose effective top-p values which generalize
>
> - **Intuitive Tuning**: Our findings indicate that min-p sampling follows a clear mathematical relationship with temperature, approximating a 1/x relationship where the optimal `p_base` approaches 1 as temperature increases. This allows for extreme but stable configurations—we successfully tested temperatures as high as 500 with min-p values of 0.994 while maintaining coherence. This pattern enables practitioners to predictably tune the creativity-coherence trade-off.
>
> - **Comparative Stability**: While all sampling methods are sensitive to hyperparameters, min-p sampling offers more predictable and controllable behavior, especially in high-temperature settings. This is evidenced by our accuracy-diversity plots, which show min-p's superior coverage of the parameter space compared to top-p's clustered behavior.
>
> **Conclusion**: We have provided practical advice to help users select appropriate hyperparameters, making min-p sampling both effective and user-friendly.
>
> ---
>
> ## 8. Novelty and Impact
>
> While sampling methods like top-p and temperature scaling are widely used, **min-p introduces a fundamentally new perspective**: sampling thresholds should dynamically adapt to model confidence. This bridges the gap between theoretical understanding of LLM sampling and practical needs for coherent and creative text generation.
>
> Our comprehensive evaluations across modern LLM benchmarks provide the first systematic study of how sampling methods affect downstream task performance at different temperature ranges. The rapid adoption of min-p by the open-source community (with integrations in VLLM, SGLang, and HuggingFace Transformers) demonstrates its practical value.
>
> By enabling coherent high-temperature sampling, min-p unlocks new capabilities like creative writing and exploring diverse reasoning paths, through a simple method with minimal added overhead.
>
> ---
>
> ## Conclusion
>
> In conclusion, we have invested substantial effort to address reviewer feedback through extensive new experiments and analyses. These enhancements significantly strengthen our paper's contributions, providing both theoretical insights into sampling dynamics and practical guidance for implementing min-p sampling. Consistent results across model families and benchmarks demonstrate min-p's broad applicability and robust performance.
>
> We hope you will consider these substantial improvements in your evaluation. Thank you again for your insightful reviews and the opportunity to strengthen this work.
>
>
> **References**
> 1. [Ari Holtzman, Jan Buys, Maxwell Forbes, Antoine Bosselut, David Golub, Yejin Choi. *The Curious Case of Neural Text Degeneration*. arXiv, 2020.](https://arxiv.org/abs/1904.09751)

---

### Public Comment · ~Nguyen_Nhat_Minh1 · 2025-03-16
**Final Camera-Ready Updates: Supplementary Human Evaluation, External Validation, and Methodology Clarifications**

Dear Area Chairs and Program Chairs,

As we prepare the camera-ready version of our paper "Turning Up the Heat: Min-p Sampling for Creative and Coherent LLM Outputs" (Submission number: 11935), we would like to transparently disclose several substantive updates we have made since the review process concluded. These changes were undertaken to improve accuracy and transparency following post-review discussions with researchers clarifying our results.

The core claims, methodology, and findings of our paper remain unchanged. However, we have made the following updates to ensure greater scientific rigor:

1. **Human Evaluation Methodology**: We conducted an additional human evaluation using the VLLM inference engine instead of Hugging Face Transformers after discovering that Hugging Face applies temperature after truncation sampling (rather than before), which reduces the effect of truncation. Other methodology improvements include using Prolific's newly introduced AI Testers feature, full-length story outputs instead of short samples to assess longform textual output coherence, more rigorous attention checks, and clearer evaluation criteria. This new evaluation, detailed in *Appendix C.2: Additional Human Evaluation with VLLM Inference Engine*, shows dramatically more pronounced advantages for min-p at high temperatures - 5/10 (min-p) vs 1/10 (top-p and baseline) for Quality and Creativity at temperature 3.0, strengthening our findings.

2. **Additional External Validation**: We added brief reference to independent EQ-Bench evaluations that further validate min-p's advantages for creative writing (scores of 62 vs baseline 51.5). Responding to prior reviewer queries regarding real-world applications, we also detail examples of inference providers that have adopted min-p or papers that have replicated our results, such as "Training a Generally Curious Agent" (Tajwar et al., 2025) [1], which specifically cites min-p's benefits for generating high-quality and diverse training data.

3. **Community Adoption Metrics Clarification**: We revised our statement about GitHub adoption to be more conservative and easily verifiable. Our original claim of "54,000 repositories and 1.1 million stars" was based on preliminary GitHub searches that included false positives. Since it is hard to exhaustively search through and verify thousands of integrations, we now report only verified integrations in the top dozen or so major frameworks (575k stars, 290k+ downstream repositories), with detailed methodology in *Appendix A.5: Detailed Community Adoption Statistics.*

These changes provide more accurate metrics, additional supporting evidence, and greater methodological transparency. We are grateful for the thorough review process that helped strengthen our work.

Sincerely,

The Authors

[1] Tajwar, F., Jiang, Y., Thankaraj, A., Rahman, S. S., Kolter, J. Z., Schneider, J., & Salakhutdinov, R. (2025). Training a Generally Curious Agent. *International Conference on Learning Representations*. Retrieved from https://openreview.net/forum?id=aC6Dc9hiu1

---

### Meta-Review · Area_Chair_5p8J · 2024-12-19

**Metareview:**

The paper provides a new sampling technique for text generation. The new technique is simple and is already widely adopted by the community (as mentioned by D38H, “The usage of it in 54,000 Github repositories alone is very impressive”). The authors provide a comprehensive analysis of their sampling technique comparing it to the previous methods, and demonstrating that in the low temperature regime, this new technique provides a significant advantage.

The reviews raised issues that are not really concerns about the paper’s quality but suggestions to improve it: Test more LLMs, evaluate creativity in additional ways, etc. In the rebuttal, the authors provided additional experiments and clarified some issues raised in the reviews, addressing almost all of the points raised in the reviews, even those that I found to be “nice-to-have".

The resulting review scores reflect the high quality of the paper: It presents convincing experiments, thorough analysis, and the provided method has an extremely high impact. The paper should be a great addition to ICLR.

**Additional Comments On Reviewer Discussion:**

see meta review

---

### Decision · Program_Chairs · 2025-01-22

Accept (Oral)